# Reproductive Effects of Exposure to Low-Dose Ionizing Radiation: A Long-Term Follow-Up of Immigrant Women Exposed to the Chernobyl Accident

**DOI:** 10.3390/jcm9061786

**Published:** 2020-06-08

**Authors:** Julie Cwikel, Ruslan Sergienko, Gil Gutvirtz, Rachel Abramovitz, Danna Slusky, Michael Quastel, Eyal Sheiner

**Affiliations:** 1The Center for Women’s Health Studies and Promotion, Ben Gurion University of the Negev, POB 653, Beer Sheva 84105, Israel; rachelabra1@gmail.com; 2Department of Epidemiology, Faculty of Health Sciences, Ben Gurion University of the Negev, POB 653, Beer Sheva 84105, Israel; sergienk@bgu.ac.il; 3OB-GYN (Women’s B ward), Soroka University Medical Center, Beer Sheva 84105, Israel; Giltzik@gmail.com (G.G.); Sheiner@bgu.ac.il (E.S.); 4Independent Researcher, San Francisco, CA, USA; danna@slusky.com; 5Nuclear Medicine, Soroka University Medical Center (retired), Beer Sheva 84105, Israel; maay100@bgu.ac.il

**Keywords:** Chernobyl, radiation, exposure, fertility, pregnancy, follow up, long term

## Abstract

The Chernobyl accident in 1986 spread ionizing radiation over extensive areas of Belarus and Ukraine, leading to adverse health effects in exposed children. More than 30 years later, exposed children have grown and became parents themselves. This retrospective study from Israel was aimed to evaluate whether Chernobyl-exposed women are at higher risk for adverse reproductive outcomes. Exposed immigrants were identified as high or low exposure based on Caesium-137 soil contamination levels registered in the town they lived in. The exposed group was age matched with three comparison groups: non-exposed immigrant women from the Former Soviet Union (FSU) excluding Belarus and the Ukraine, immigrants from other countries (Non FSU) and Israeli-born women at a ratio of 1:10. Chernobyl-exposed women were more likely to be nulliparous and have fewer children (2.1 + 0.8 vs. 3.1 + 1.8, *p* < 0.001), were more likely to undergo fertility treatments (8.8% vs. 5.8%, adjusted OR = 1.8, 95%CI 1.04–3.2, *p* = 0.036), and were also more likely to have anemia after delivery (49.4% vs. 36.6%, OR = 1.7, 95%CI 1.2–2.3, *p* = 0.001), compared to women in the combined comparison groups. The overall fertility of Chernobyl-exposed women seems to be reduced as reflected by the lower number of children and their greater need for fertility treatments.

## 1. Introduction

The worst environmental disaster of the twentieth century occurred in the early morning of 26th, April 1986, when the nuclear reactor at Chernobyl exploded blowing off the 1000-ton concrete roof and dispersing radioactive particles all over Europe. The explosion caused the graphite outer layer of the reactor to burn for days and firemen, soldiers, students and volunteers were brought in from all over the Soviet Union to try to “liquidate” the burning, giving rise to the term “liquidators” for the highly exposed salvage workers. Approximately 135,000 persons were evacuated from their homes with no warning in the days following the accident, especially from the “company town” of Pripyat and in the 30km radius around the damaged reactor [1].

Contamination occurred through different pathways: (1) direct exposure of body surfaces to radioactive strontium, plutonium, iodine and cesium from the reactor core, with each element tending to aggregate in a different organ in the body, e.g., radioactive iodine is absorbed in the thyroid, radioactive cesium in the soft tissues [2], (2) inhalation of radioactive dust; and (3) indirect absorption of radioactive, contaminated food and water, caused by contamination of the water sources and the food chain. Most of the civilian populations were exposed to multiple pathways over time [1,2]. Cancer, primarily thyroid cancer, has been documented among exposed cohorts [3,4].

A majority of those residing in the most contaminated areas wished to relocate to a different part of the Soviet Union or leave the country [5]. Close to 200,000 of the exposed population immigrated to Israel starting in 1989, wanting to distance themselves from the contaminated areas, and seeking treatment for “exposure to radiation” [6]. Israel, under the Law of Return, offered full access to universal health care on arrival and therefore these immigrants had access to health care comparable to all other Israeli citizens and their health data were integrated into the Israeli Health Fund systems.

In 1990, responding to the requests of Chernobyl-exposed immigrants for health assessments at the peak of immigration from the FSU, a clinic was established by Prof. Michael Quastel at Soroka University Medical Center (SUMC) in association with the Faculty of Health Sciences of Ben-Gurion University of the Negev (BGU) to evaluate the health of this population [7]. Whole body counts of cesium (Cs137), physical examinations and questionnaire data were collected in the framework of the Ben Gurion University Chernobyl Study (BGUCS) from a voluntary sample of 723 adults and 650 children under the age of 18 [7,8,9,10]. In 1994 and with a two-year follow-up in 1996, additional BGUCS research was conducted on psychosocial and self-reported health indicators of Chernobyl-exposed immigrants (from high and low contamination areas) compared with non-exposed immigrants from the FSU. Higher prevalence of stress as indicated by post-traumatic stress disorder (PTSD), depression, anxiety, somatization and measured blood pressure were found among high exposed groups compared with low exposed groups [11,12,13]. In 2004, Slusky et al. examined the medical and mortality records of 1447 of the original BGUCS cohort and found an increased odds ratio (OR) of respiratory disorders, a borderline increase in ischemic heart disease and a greater number of visits to primary care physicians among the high exposed compared to the low exposed Chernobyl immigrants [14].

### A Review of Reproductive Outcomes Among Chernobyl-Exposed Populations

One major concern among Chernobyl-exposed populations, was worry over reproductive health, congenital malformations and genetic damage among offspring [4,15,16,17,18,19,20]. The anticipated excess increase in stillbirths and congenital malformations was not found in all studies of Chernobyl-exposed women and children [4,17,18,21,22]. However, a recent review concluded that genetic damage was evident in some of the exposed populations, particularly if the parents were exposed to high doses stemming from on-site exposure among liquidators or if the children continued to be exposed over many years [4].

Studies regarding the reproductive health of women in the contaminated regions immediately after the Chernobyl accident showed a decrease in birth rates, an increase in anemia during pregnancy, and an increase in perinatal mortality. In 1987, the highest recorded level of perinatal mortality was 37.4% in the Polessky district of the Ukraine, when the stillbirth and early neonatal mortality rates were 20.6% and 16.8%, respectively [23,24,25]. Accompanying the increase in morbidity and neonatal mortality were high rates of anxiety and depression in the contaminated regions among women of reproductive ages [15,21,26]. Worry over health issues was compounded by economic stress and loss, estimated as a 7% decrease in GNP associated with the Chernobyl disaster in the Ukraine [27].

According to a study by Rachmatolin et al., following the reports of the radiation contamination in the region, many women opted to induce abortions, with an increase as high as 275% over pre-accident rates [28]. Pregnant women in the contaminated areas around Chernobyl were encouraged to abort and many did so [15]. At the time, clinicians relied on what had been learned from the populations exposed to the atomic bombs at Hiroshima and Nagasaki, which showed that damage to the fetus was most likely during the 8–25th week of gestation and on conventional wisdom derived from occupationally exposed cohorts [29,30,31]. In hindsight, some feel that misconceived fear over potential congenital malformations led to thousands of needlessly induced abortions [32,33].

Significant reproductive effects were apparent also in countries of Europe that were contaminated by the radioactive plume that drifted from the crippled reactor. For example, in Denmark, many women, alarmed over the reports of radiation reaching their countries, opted for induced abortions in the three months following the accident [32], while in Greece 23% of early pregnancies were aborted [34]. In Finland and Norway, an increase in spontaneous abortions following the accident was noted, while the rates of induced abortion and stillbirth did not reflect the trends reported in other countries in Europe [35,36]. Several countries (Italy, Norway, Sweden) reported a significant decrease in conception rates in the months following the accident as couples apparently postponed having children for fear of adverse health effects among their offspring [36,37,38].

These reproductive decreases were paralleled by increases in perinatal mortality reported in 1987–1990 in Germany and Poland, attributed to late effects of strontium exposure [24]. Increases in mental retardation and low IQ were detected among children exposed in utero whose mothers had been evacuated from Pripyat compared to their classmates from Kiev [39].

Prof. Maureen Hatch and colleagues evaluated the effect of fetal exposure to radioactive iodine (I^131^) through a retrospective review of a Ukrainian cohort of 2582 exposed in utero. This analysis showed a dose-dependent reduction in head and chest circumference with a fetal dose dependent increase in gestational length (0.5 weeks/Gy) but not an increase in birth weight. They concluded that the observed effects were consistent with radiation effects among utero-exposed infants of Japanese atomic bomb survivors [40]. A team led by Prof. Eugenia Stepanova in Kiev examined the health outcomes of intra-uterine irradiation associated with Chernobyl exposure among mothers with acute and chronic exposure and compared them with a control group of children presumably non-exposed. Cytogenic studies using G-banding were conducted with a sub-sample of 57 children. Developmental anomalies were correlated with radiation dose with the threshold dose around 0.75 Gy. The number of anomalies was inversely correlated with the gestational age at the time of the accident [41].

Summarizing the research on reproductive health effects among mothers exposed to radiation associated with the Chernobyl accident shows that there were significant effects, partly mitigated by significant fears about the health effects of exposure on fetal and child health. However, the research findings are often inconsistent, and few long term health effects have been studied among the cohorts of children who are now women of reproductive age themselves [4,11,14,42].

The aim of this study was to characterize the reproductive history of women who have been exposed to ionizing radiation from Chernobyl as children or young adults and to compare their long-term reproductive health status to the reproductive health status of three comparison groups: non-exposed immigrants from the former Soviet Union (FSU), immigrants from other countries, and native born Israelis.

## 2. Experimental Section

Capitalizing on several unique data sets from Quastel, Cwikel and Slusky [7,14,43], we traced Chernobyl-exposed women who were children or young adults at the time of the Chernobyl accident and subsequently immigrated to Israel and had at least one child at SUMC (*n* = 170). SUMC is the only hospital serving the Negev region, where many of the Chernobyl-exposed immigrants settled. Using record linkage techniques that identify each citizen with a unique identity number, we assessed the reproductive histories of exposed and non-exposed women in order to estimate rates of Unfavorable Reproductive Outcomes (UROs) [30].

Three comparison groups of women were constructed and up to 10 cases of women were randomly selected from the SUMC records, matched on age within 1 year: (1) immigrants from the FSU who did not originate in areas affected by Chernobyl, i.e., not Ukraine and Belarus; (2) immigrants to Israel from other countries (not FSU) (Europe, North and South America, North Africa and countries of the Middle East); and (3) native-born Israeli women (excluding native-born Israeli Bedouin women whose health profile is significantly different from other Israeli-born populations [44]). We combined data sets to produce a set of 3652 women who had given birth at least once at SUMC and whose data were included in the perinatal database. Births with multiple gestations were excluded.

The perinatal database from SUMC contains background information on the birthing mothers: ethnic group and country of immigration, date of immigration, smoking status, BMI and age [45,46,47]. The SUMC perinatal database also includes significant URO variables derived from ICD-Codes for complications of pregnancy, childbirth and the puerperium (Codes 630–677). In this study we included:(1)Background reproductive health and history–age, smoking, obesity (defined as BMI ≥ 30 kg/m^2^), chronic hypertension, diabetes mellitus (pre-gestational), recurrent pregnancy loss, fertility treatments including all assisted reproductive techniques (ovulation induction, intrauterine insemination and in-vitro fertilization)(2)Pregnancy complications-hypertensive disorders of pregnancy (gestational hypertension, pre-eclampsia and eclampsia), gestational diabetes, intrauterine growth restriction (IUGR)(3)Delivery complications-premature rupture of membranes, meconium stained amniotic fluid, pathological presentation, prolonged first or second stage of labor, placental previa, placental abruption, malpresentation, postpartum hemorrhage, cesarean section, uterine rupture, blood transfusion(4)Adverse birth outcomes-malformations, low Apgar scores (<7) at 5 min, perinatal mortality including: stillbirth, intrapartum death -IPD, postpartum death- PPD (up to 30 days post-partum), anemia (<10.0 hemoglobin), preterm delivery (<37 weeks), and macrosomia (birthweight > 4000 g) [45,46,47].

### 2.1. Ethical Considerations

Data were analyzed from SUMC records without any personal contact or interviews. Furthermore, the resulting data set was prepared for data analysis without identifying information such as name and identity number. Thus, direct, current, informed consent was not required. At the time of study entry into the original cohort, respondents signed informed consent forms and their Israeli identity numbers were recorded. All of the researchers in the project who were involved in the data analysis have completed the online Medical Ethics in Research training. The research protocol received Helsinki approval from the SUMC committee.

### 2.2. Data Collection

This retrospective cohort study is based on data sets initially collected in 1991-1994, with the removal of duplicate cases who appeared in more than one data set. Using the International Atomic Energy Agency (IAEC) maps prepared in the year after the accident, the immigrants were divided into two groups based on the 137Cs soil contamination levels registered in the town they lived in just prior to immigration to Israel [48]. The term ‘high exposure’ was defined as contaminated with ≥37 GBq/km^2^ and “low exposure” was defined as less than this level of contamination. Most of the high exposure individuals were from Gomel and towns in this area and low exposure respondents were from Kiev, Minsk and surrounding towns.

In the subsample used for this study, the women who had ever given birth at SUMC were extracted from the sample of Chernobyl-exposed immigrants (*N* = 1128 or 92.4% of the Slusky data set) and then their data were cross-linked through the computerized perinatal hospitalization data set of SUMC from 1992 (by which date the bulk of the FSU immigration had arrived) to the end of 2017. Thus, the data presented here are all hospitalizations related to reproductive health at one hospital (SUMC), which serves the southern region of Israel.

### 2.3. Statistical Analyses

Statistical analyses were performed using the SPSS package (version 25.0) (SPSS, Chicago, IL, USA). Statistical significance of categorical data was tested with chi-square, Fisher’s exact test and Scheffé post-hoc comparisons derived from ANOVA. For continuous variables, F-statistics were used. Logistic regression models were constructed to control for confounders.

## 3. Results

Table 1 and Table 2 show the background features and obstetric outcomes of the five groups: high and low Chernobyl-exposed, FSU control, immigrant control and Israeli groups of women. Chernobyl-exposed women were more likely to be nulliparous at the index birth (27.4% and 22.7% for high and low exposure groups, respectively), compared with FSU immigrants (17.5%) and other immigrants (13.3%) and Israelis (13.5%). The Chernobyl-exposed groups were quite unlikely to have four and more children (4.1% and 5.2%, high and low exposure, respectively) while this was much more common among other FSU immigrants (18.6%), other immigrants (41.3%) and Israeli women (38.3%).

These findings are congruent with the fact that the lowest overall fertility was apparent among the Chernobyl-exposed groups (according to the maximal birth number): (2.0), which is considerably lower than the FSU control (2.6) and the Non-FSU immigrant and Israel born groups (3.4 and 3.3, respectively). In general, both the Chernobyl-exposed and the FSU controls have smaller families than the other comparison groups. However, in a sub-analysis, comparing all Chernobyl-exposed with all non-exposed women, Chernobyl-exposed women had fewer children (2.0 + 0.8 vs. 3.1 + 1.8, *p* < 0.001). While the majority of all groups were married, the exposed groups were the least likely to be married, and the high exposure group had the highest portion of women who were single mothers (19.4%) compared to any other group (5.6%, 8.8%, 4.0%, 5.4%, 6.5%, low exposure, FSU, other immigrants, Israel born, respectively, χ^2^ = 69.178, d.f. 16, *p* < 0.001).

Perinatal outcomes are presented in Table 3 and, overall, were comparable between the groups. However, exposed women compared to women from the combined comparison groups were more likely to undergo fertility treatments (8.8% vs. 5.8%, adjusted OR for age at delivery using multiple logistic regression = 1.8, 95%CI 1.04-3.2, *p* = 0.036), and were also more likely to have anemia after delivery (49.4% vs. 36.6%, adjusted OR for age at delivery = 1.7, 95%CI 1.2–2.3, *p* = 0.001). However, they had fewer infant malformations (4.7% vs. 9.1%, OR = 0.975, 95%CI 0.970–0.994, *p* = 0.026) and less meconium staining (13.5% vs. 22.2%, OR = 0.977, 95%CI 0.962–0.991, *p* = 0.003). In Table 3 we see that while the difference did not reach statistical significance, the high exposure group had a very low rate of smoking (1.4%) and there were no observed cases of obesity (0.0) compared with the other non-exposed groups.

## 4. Discussion

This study investigated the reproductive outcomes of women who were exposed to the Chernobyl nuclear plant accident as children or young adults. The most striking finding from this cohort of women is their reduced fertility that is expressed by significantly smaller families, higher rates of nulliparity at the index birth and a greater probability that they needed to receive fertility treatment in order to succeed in bringing an infant into the world. It should be noted that Israel has a very liberal policy regarding IVF treatments: they are fully subsidized with unlimited cycles up until two live births have been achieved, covering women up until the age of 45. Thus, the women in this study had access to a full range of fertility treatment options [49].

This finding of reduced fertility contradicts the latest assessment from the World Health Organization. In 2006, the Report of the UN Chernobyl Forum Expert Group “Health” concluded that “Given the low radiation doses received by most people exposed to the Chernobyl accident, no effects on fertility, numbers of stillbirths, adverse pregnancy outcomes or delivery complications have been demonstrated nor are there expected to be any” [50]. Of course, radiation exposure can affect male fertility as well as female [2]. It is possible that women affected by Chernobyl also married partners who were themselves exposed, either to Chernobyl-associated radiation sources or to other occupational radiation exposures, further compromising fertility. Furthermore, a more complete picture of women’s fertility can be inferred from age of menarche and menstrual cyclicity, however these data were not available to us. In future studies it would be appropriate to gather data on these variables as well as exposure histories of both parents, particularly for populations still living in the contaminated regions.

While this study reaffirms that no adverse pregnancy or delivery complications were found among women exposed to the Chernobyl accident compared to women in the comparison groups, their fertility was negatively affected. Previous research shows that radiation exposure is damaging to ovarian tissue with effects that are dose and age dependent, and particularly toxic to oocytes [51]. A study of ovarian function after radiation, showed that 27 of 38 patients who received whole abdominal irradiation in childhood failed to undergo complete pubertal development, and 10 later developed premature menopause. All had elevated FSH levels and low estradiol levels [52,53]. Data from female cancer survivors show that the chance of pregnancy is lower and the risk of non-surgical premature menopause is increased after radiation exposure. Although the risk was most elevated among those exposed to the highest doses of ovarian irradiation (≥ 1000 cGy), even exposure to doses as low as 1 to 99 cGy were associated with an increased risk of nonsurgical premature menopause compared with cancer survivors who received no radiation [54]. The Chernobyl-exposed women are believed to have been exposed to radiation and radioactive toxins through multiple pathways, which may all lead to ovarian dysfunction and present, over time, as reduced fertility [55].

As mentioned, another important finding of this study is that once the Chernobyl-exposed women achieved pregnancy, its course and outcome was comparable to the comparison groups. Although some publications on the Chernobyl accident found increased prevalence of congenital malformations in exposed population [25], our study found a comparable incidence of congenital malformations in exposed women and the comparison groups. This is in accordance to the conclusion published by the UNSCEAR in 2001, after an extensive review of the literature, that stated that no radiation induced birth defects or congenital malformations have been observed in humans following the accident [55]. This conclusion was also reinforced in the UN health report in 2006, stating that “the Chernobyl epidemiological studies do not indicate a radiation related increase in malformations as a direct result of radiation exposure” [50].

Regarding pregnancy or delivery complications, Little (1993) provided a comprehensive review of all the Chernobyl studies related to adverse reproductive outcomes and pointed out that there was no consistent evidence of measurable adverse outcomes of pregnancy (including miscarriages, perinatal mortality, low birth weight), but there was an increase in indirect effects such as abortions and lower birth rate, presumably related to anxiety [56]. The UN health report (2006) on this topic was more cautious, stating that “The descriptive nature of information provided to the Expert Group on stillbirths and adverse complications of pregnancy has prevented any conclusion regarding radiation effects”. However, the report did mention that “prior non-Chernobyl scientific literature at these dose levels would not support a radiation related effect. Further, we have not seen adequate scientific data clearly indicating an effect”. It seems logical to assume that if no adverse pregnancy outcomes were noted in those exposed during pregnancy at the time of the accident, there would be no reason for these complications to effect later pregnancies of our cohort.

The finding that the exposed women are more likely to have anemia may represent a late effect of radiation exposure. A study of pregnant women from heavily contaminated areas of Belarus, comparing morbidity from before the accident to after, found high rates of anemia among both mothers and neonates following the accident [57]. Other studies comparing women from high exposure areas (e.g., Zhitomir) with low exposure areas (e.g., Kiev) found that there was a dose response gradient for higher exposed mothers on perinatal anemia [17,25]. We too found an exposure gradient in our data, with the highest rates of post-partum anemia among the high exposure group, a significantly greater excess among the low-exposed and both exposed groups significantly higher than the control groups.

Another interesting finding that is apparent in these data is the low level of smoking among the high exposed group in comparison to all the other groups. According to the model proposed by Cwikel, Gidron and Quastel [58] that outlines the various effects of low level of radiation on the health of populations, one aspect of being exposed to environmental hazards is the change that may be apparent in health behaviors. Thus, it is possible that women who understand that they have been exposed, will try to mitigate their risks by controlling those behaviors under their control to reduce other sources of risk, e.g., by not smoking and not being overweight. A very low level of smoking among highly exposed mothers compared to the comparison group was observed also in the cohort followed by Bromet and colleagues [59].

## 5. Conclusions

The overall fertility of Chernobyl-exposed women seems to be reduced as reflected by the lower number of children and their greater need for fertility treatments. Our findings raise concerns regarding the long-term implications of the Chernobyl disaster.

## Figures and Tables

**Table 1 jcm-09-01786-t001:** Demographics and clinical characteristics of the study groups.

	Exposed Sub-Groups (%)		Non-Exposed Sub-Groups (%)				
	High Exposure (*n* = 73)	Low Exposure (*n* = 97)	FSU Immigrants (*n* = 1427)	Non-FSU Immigrants (*n* = 1486)	Israeli (*n* = 569)	Total (3652)	*P* Value
**Mother age at last birth (± SD)**	31.1 ± 4.8	30.9 ± 4.5	30.4 ± 5.2	33.4 ± 5.2	33.0 ± 5.1		<0.001
**Max. birth number**	2.0 ± 0.8	2.1 ± 0.9	2.6 ± 1.3	3.4 ± 1.9	3.3 ± 1.8		<0.001
**Age at accident** **<17** **18-25** **26-35** **35+**	72.6 19.2 8.2 0.0	72.2 23.7 4.1 0.0	68.1 19.5 11.8 0.6	45.6 25.2 26.4 2.8	48.0 17.2 31.1 3.7	56.0 21.6 20.5 1.9	<0.01
**Parity** **1** **2-3** **4+**	27.4 68.5 4.1	22.7 72.2 5.2	17.5 63.8 18.6	13.3 45.5 41.3	13.5 48.2 38.3	15.5 54.2 30.3	<0.001
**Smoking**	1.4	5.2	5.7	4.2	6.0	5.0	0.140
**Blood transfusion**	1.4	0.0	1.9	2.7	1.6	2.1	0.213
**Obesity**	0.0	2.1	1.9	2.0	1.9	1.9	0.834
**Diabetes mellitus**	19.2	12.4	14.6	12.8	13.5	13.7	0.390
**Recurrent miscarriage**	4.1	4.1	5.0	7.7	8.1	6.6	<0.01
**Fertility treatments**	8.2	9.3	5.4	5.9	6.5	5.9	0.449
**Preterm delivery < 37**	11.0	17.5	13.4	12.8	13.5	13.2	0.695

**Table 2 jcm-09-01786-t002:** Obstetric outcomes of the study groups.

	Exposed Sub-Groups (%)		Non-Exposed Sub-Groups (%)				
	High Exposure (*n* = 73)	Low Exposure (*n* = 97)	FSU Immigrants (*n* = 1427)	Non-FSU Immigrants (*n* = 1486)	Israeli (*n* = 569)	Total (3652)	*P* Value
**Caesarian Section**	30.1	23.7	23.0	21.3	24.3	22.6	0.292
**Gestational Diabetes**	17.8	7.2	12.1	11.4	12.1	11.8	0.302
**Any Diabetes**	19.2	12.4	14.6	12.8	13.5	13.7	0.390
**IUGR**	5.5	5.2	4.0	3.9	3.9	4.0	0.936
**Placenta previa**	0.0	1.0	1.0	1.3	1.4	1.2	0.779
**Pathological presentation**	9.6	8.2	9.7	10.2	10.7	10.0	0.921
**Premature rupture of membranes (PROM)**	15.1	17.5	17.2	14.6	16.5	16.0	0.414
**Mild preeclampsia**	8.2	7.2	8.3	7.6	5.6	7.6	0.387
**Severe preeclampsia**	0.0	2.1	2.4	2.2	2.1	2.2	0.758
**Hypertensive disorders**	13.7	7.2	12.8	10.9	9.7	11.4	0.152
**Non progressive labor 1^st^ stage**	5.5	6.2	3.4	3.9	3.7	3.8	0.615
**Non progressive labor 2^nd^ stage**	4.1	2.1	2.2	2.7	2.3	2.4	0.773
**Placental abruption**	1.4	1.0	0.9	1.1	1.8	1.2	0.625
**Induction of labor**	30.1	33.0	23.8	22.1	22.1	23.2	0.067
**Uterine rupture**	0.0	0.0	0.1	0.3	0.2	0.2	0.910
**Post-partum hemorrhage**	1.4	1.0	1.0	1.5	0.9	1.2	0.624

**Table 3 jcm-09-01786-t003:** Perinatal outcomes of the study groups.

	Exposed Sub-Groups (%)		Non-Exposed Sub-Groups (%)				
	High Exposure (*n* = 73)	Low Exposure (*n* = 97)	FSU Immigrants (*n* = 1427)	Non-FSU Immigrants (*n* = 1486)	Israeli (*n* = 569)	Total (3652)	*P* Value
**Low Birth Weight ≤2500**	11.0	12.4	11.0	13.8	13.0	12.5	0.239
**Macrosomia**	6.8	8.2	9.1	6.8	8.3	8.0	0.238
**Anemia**	54.8	45.4	39.7	34.9	33.7	37.2	<0.01
**Congenital malformations**	6.8	3.1	9.0	9.2	9.3	8.9	0.318
**Apgar 5 minutes < 7**	0.0	0.0	0.6	1.3	0.4	0.8	0.114
**Perinatal mortality**	1.4	2.1	2.0	2.3	1.9	2.1	0.957
**Stillbirth**	1.4	1.0	1.1	1.3	1.2	1.2	0.995
**Intrapartum death**	0.0	0.0	0.2	0.1	0.0	0.1	0.813
**Post-partum death**	0.0	1.0	0.6	1.0	0.7	0.8	0.723

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
