# Peer review of "Reproductive Effects of Exposure to Low-Dose Ionizing Radiation: A Long-Term Follow-Up of Immigrant Women Exposed to the Chernobyl Accident"

_jcm, 2020, doi:10.3390/jcm9061786_

Round 1

Reviewer 1 Report

This is a very interesting, almost review of the health effects of people - women - who were exposed to the Chernobyl accident.

The review and background in the introduction is absolutely captivating. It is well written and flows very well.

The study design is very descriptive. The results are not overstated. However, the age of nulliparous pregnancy is very young in the exposed group. Whether that is related to exposure or related to age is not adequately explained. My real question is nulliparous data significant if controlled for age. I realize this was age matched but it seems the ages were very different.

Was there an association with degree of anemia (continuous variable) and exposure (High versus low) or exposure versus not exposed?

In terms of fertility treatment, does Isreal with its health system provide for fertility treatments in a systematic step-wise approach? A caveate is that fertility takes male and female.

Author Response

1. This is a very interesting, almost review of the health effects of people - women - who were exposed to the Chernobyl accident. The review and background in the introduction is absolutely captivating. It is well written and flows very well.

Answer: We very much appreciate that you found the introduction interesting, well written and the background captivating.  Given that it is now 33 years since the accident, there are many clinicians working today who have little idea as to the events that happened at Chernobyl which is why we found it important to give the context for the study.

2. The study design is very descriptive. The results are not overstated. However, the age of nulliparous pregnancy is very young in the exposed group. Whether that is related to exposure or related to age is not adequately explained. My real question is nulliparous data significant if controlled for age. I realize this was age matched but it seems the ages were very different.

Answer: We thank the reviewer for the comment; the age is presented as age at the accident. However, the age differences at the index birth are non-clinically significant: 31 in the high exposure, 30.9 in the low, 30.4 in the control FSU immigrants, and 33 in the other controls. Nulliparity is associated with the fact that these women had a lower number of births. Indeed, Chernobyl-exposed women had fewer children (2.0+0.8 vs. 3.1+1.8, p<.0001).

3. Was there an association with degree of anemia (continuous variable) and exposure (High versus low) or exposure versus not exposed?

Answer: Unfortunately, we do not have data on anemia as a continuous variable. We have defined this variable in the Methods section as anemia (< 10.0 hemoglobin).

4. In terms of fertility treatment, does Israel with its health system provide for fertility treatments in a systematic step-wise approach? A caveat is that fertility takes male and female.

Answer: We agree with the reviewer that fertility involves both male and female reproductive capacity and have added this comment to the discussion (see lines 258-265).

We also added a brief overview of the Israeli system of fertility treatment and eligibility to make the context clear (see lines 250-253). Unfortunately, we do not have data regarding the cause of infertility. This was added to the discussion section as a limitation of the study (see lines 258-265).

Reviewer 2 Report

The review manuscript, ’Reproductive effects of exposure to low-dose ionizing radiation: a long-term follow-up of immigrant women exposed to the Chernobyl accident’ by Cwikel J, et al showed that the fertility from radiation-exposed women decreased, supported by the lower number of children and the higher number of fertility treatment. While the results are different from WHO data, this data is comparable with the data from the populations exposed to the atomic bombs at Hiroshima and Nagasaki, supported with the study of radiation effect on reproduction. This is a very important result and clearly described. Here are minor comments to edit.

Minor comments

  1. In line 49, put the references for this sentence (radioactive iodine is absorbed in the thyroid, radioactive cesium in the soft tissues.
  2. If the authors have data about the average age of menarche and period of menstrual cyclicity from the same groups for Table 1, include or mention those in the results or discussion part.
  1. In line 266, put a period after the sentence.
  2. In References #2 and #52, the authors should delete ‘(accessed on)’.

Author Response

The review manuscript, ’Reproductive effects of exposure to low-dose ionizing radiation: a long-term follow-up of immigrant women exposed to the Chernobyl accident’ by Cwikel J, et al showed that the fertility from radiation-exposed women decreased, supported by the lower number of children and the higher number of fertility treatment. While the results are different from WHO data, this data is comparable with the data from the populations exposed to the atomic bombs at Hiroshima and Nagasaki, supported with the study of radiation effect on reproduction. This is a very important result and clearly described. Here are minor comments to edit.

We thank the reviewer for the appreciative comments.  

Minor comments

  1. In line 49, put the references for this sentence (radioactive iodine is absorbed in the thyroid, radioactive cesium in the soft tissues.

Answer: The reference was added accordingly.

  1. If the authors have data about the average age of menarche and period of menstrual cyclicity from the same groups for Table 1, include or mention those in the results or discussion part.

Answer: Unfortunately we do not have data about age of menarche, so we have added this as a limitation in the discussion (see lines 258-265)

  1. In line 266, put a period after the sentence.

Answer: Done.

  1. In References #2 and #52, the authors should delete ‘(accessed on)’.

Answer: Done.